Manuscript prepared for Biogeosciences
with version 2014/09/16 7.15 Copernicus papers of the LaTeX class copernicus.cls.
Date: 1 December 2017

# Carbon-climate feedbacks accelerate ocean acidification

Richard J. Matear[1] and Andrew Lenton[1,2]

[1]CSIRO Oceans and Atmosphere, Hobart, Tasmania, Australia
[2]Antarctic Climate and Ecosystems CRC, Hobart, Tasmania, Australia

*Correspondence to:* Richard Matear
(richard.matear@csiro.au)

**Abstract.** Carbon-climate feedbacks have the potential to significantly impact the future climate by altering atmospheric $CO_2$ concentrations (Zaehle et al., 2010). By modifying the future atmospheric $CO_2$ concentrations, the carbon-climate feedbacks will also influence the future ocean acidification trajectory. Here, we use the $CO_2$ emissions scenarios from four Representative Concentration Pathways (RCPs) with an Earth System Model to project the future trajectories of ocean acidification with the inclusion of carbon-climate feedbacks. We show that simulated carbon-climate feedbacks can significantly impact the onset of under-saturated aragonite conditions in the Southern and Arctic Oceans, the suitable habitat for tropical coral and the deepwater saturation states. Under the high emission scenarios (RCP8.5 and RCP6), the carbon-climate feedbacks advance the onset of surface water under saturation and the decline in suitable coral reef habitat by a decade or more. The impacts of the carbon-climate feedbacks are most significant for the medium (RCP4.5) and low emission (RCP2.6) scenarios. For the RCP4.5 scenario, by 2100 the carbon-climate feedbacks nearly double the area of surface water under-saturated with respect to aragonite and reduce by 50% the surface water suitable for coral reefs. For the RCP2.6 scenario, by 2100 the carbon-climate feedbacks reduce the area suitable for coral reefs by 40% and increase the area of under-saturated surface water by 20%. The sensitivity of ocean acidification to the carbon-climate feedbacks in the low to medium emission scenarios is important because recent $CO_2$ emission reduction commitments are trying to transition emissions to such a scenario. Our study highlights the need to better characterise the carbon-climate feedbacks and ensure we do not underestimate the projected ocean acidification.

## 1 Introduction

Ocean acidification, the measurable consequence of increasing atmospheric $CO_2$ concentrations, has the potential to significantly impact individual marine organisms and ecosystems by reducing calcification rates (Stojkovic et al., 2013), altering phytoplankton composition (Lohbeck et al., 2012), changing fish behavior (Munday et al., 2009) and affecting larval recruitment (Ross et al., 2011). This has the potential to significantly impact the ecosystem services that the ocean provides (Gattuso et al., 2015). Therefore, accurate projections of ocean acidification are essential to assessing the future impact of ocean acidification, setting policy that avoids or limits dangerous climate change, managing marine resources, and guiding adaptation strategies.

Future carbon-climate projections generally show global warming alters the efficiency of carbon dioxide ($CO_2$) uptake by both the land and ocean (Friedlingstein et al., 2006; Roy et al., 2011; Arora et al., 2013). The land feedbacks include the influence of warming, elevated $CO_2$, and changes in soil moisture on net primary productivity and soil respiration (e.g. Friedlingstein et al. (2014)). In the ocean, the feedbacks include alterations to the ocean carbon cycle and the uptake of anthropogenic carbon from the atmosphere as a result of warming and changes in upper ocean stratification and circulation (e.g. Matear and Hirst (1999)). As a result, more emitted carbon stays in the atmosphere leading to additional warming (Friedlingstein et al., 2003, 2001), which represents a positive climate feedback. While the future carbon-climate feedbacks under the various emission scenarios are highly uncertain (Zhang et al., 2014; Wenzel et al., 2014), the carbon-climate feedbacks have the potential to significantly impact future climate trajectories (Zaehle et al., 2010). Studies show that the carbon-climate feedbacks are primarily due to changes in land carbon uptake, with large inter-model variability in how the land carbon uptake responds to the future climate (Friedlingstein et al., 2014; Arora et al., 2013). Importantly, these carbon-climate feedbacks will also influence the future trajectory of ocean acidification because the surface ocean carbon tracks the atmospheric $CO_2$ (McNeil and Matear, 2008). Therefore, the carbon-climate feedbacks are not only important to future climate change, they are also relevant to the future trajectory of ocean acidification and this study investigates whether future carbon-climate feedbacks can have important consequences for ocean acidification. This is important as these feedbacks have not been accounted for in studies that project future change in ocean acidification (Bopp et al., 2013). To address this gap in ocean acidification research, this study investigates the potential for the carbon-climate feedbacks to alter the future evolution of ocean acidification by using a global Earth System Model (ESM) (Phipps et al., 2011; Zhang et al., 2014).

For this study, we consider future projections of atmospheric $CO_2$ from four Representative Concentration Pathways (RCPs) as provided by the Coupled Model Intercomparison Project Phase 5 (CMIP5) [http://cmip-pcmdi.llnl.gov/cmip5/] based on both prescribed atmospheric $CO_2$ concentrations and emissions. The scenarios represent the high (RCP8.5, RCP 6.0), medium (RCP4.5) and low (RCP2.6) atmospheric $CO_2$ concentration pathways of the IPCC's Fifth Assessment Report (Stocker

et al., 2013). We focus our analysis on how the simulated carbon-climate feedbacks influence future ocean acidification.

The structure of the paper is as follows. In the next section, we briefly describe the ESM used and the simulations performed. In the subsequent section, we present the results from the historical and the future simulations. We show the carbon-climate feedbacks accelerate ocean acidification in all future emissions scenarios. Importantly, it is in the low and medium emissions scenarios where ocean acidification is most impacted by the carbon-climate feedbacks. For the low and medium emissions scenarios, ocean acidification is sensitive to the additional $CO_2$ in the atmosphere provided by the carbon-climate feedbacks. This has important policy relevance because the recent global commitments to reduce greenhouse gas emissions seek to put us on the low to medium emissions path to avoid dangerous climate change, but it may underestimate the consequences for ocean acidification.

## 2    Model Description

In this study, we used the CSIRO Mk3L Carbon Ocean, Atmosphere, Land (COAL) ESM (Buchanan et al., 2016) The COAL components include ocean and land biogeochemistry (Matear and Lenton, 2014; Zhang et al., 2014), which exchange $CO_2$ with the atmosphere and enable investigation of carbon-climate interactions within an ESM. The atmospheric resolution is $5.6°$ by $3.2°$, and 18 vertical layers, with the land carbon component having the same horizontal resolution as the atmosphere.

The land module (CABLE) with CASA-CNP (Wang et al., 2010; Mao et al., 2011) simulates the temporal evolution of heat, water and momentum fluxes at the surface, as well as the biogeochemical cycles of carbon, nitrogen and phosphorus in plants and soils. For this study, we use the land module that includes carbon, nitrogen and phosphate cycles with spatially explicit estimates of nitrogen deposition from Dentener (2006), which do not change with time. The simulated (Zhang et al., 2014) geographic variations of nutrient limitation, and major biogeochemical fluxes and pools on the land under the present climate conditions are consistent with published studies (Wang et al., 2010; Hedin, 2004).

The ocean component of the ESM has a resolution of $2.8°$ by $1.6°$, and 21 vertical levels. The ocean biogeochemistry is based on Matear and Hirst (2003) and Buchanan et al. (2016), and simulates the evolution of phosphate, oxygen, dissolved inorganic carbon and alkalinity in the ocean. This ocean biogeochemical model was shown to simulate realistically the global ocean oxygen and phosphate cycles (Duteil et al., 2012) and the present-day distribution of dissolved inorganic carbon and alkalinity in the ocean (Matear and Lenton, 2014; Buchanan et al., 2016). The simulations presented here use the standard ocean Biogeochemical formulation presented in Matear and Lenton (2014).

### 2.1 Model simulations.

The ESM was spun-up under pre-industrial atmospheric $CO_2$ (1850: 284.7 ppm) until the simulated climate was stable (2000 years) (Phipps et al., 2012). Stability was defined as the point where the linear trend of global mean surface temperature change over the last 400 years of the spin-up was less than $0.015°C$ per century. The ocean and land carbon cycles were then spun-up off-line using separate ocean and land simulations, using the pre-industrial climate state of Phipps et al. (2012) and the preindustrial atmospheric $CO_2$ until the drift in the global carbon exchanges was less than 0.1 PgC/ per century. Finally, the climate and carbon states obtained above were incorporated in an ESM simulation that continued for another 1000 years to ensure that the global drift in the climate and carbon were less than $0.015°C$ and 0.1 PgC per century.

From the spun-up initial climate and carbon state, the historical simulation (1850 - 2005) was performed using the historical atmospheric $CO_2$ concentrations as prescribed by the CMIP5 simulation protocol. For the historical period, the atmospheric $CO_2$ affects both the radiative properties of the atmosphere and the carbon cycle (Zhang et al., 2014). From year 2006, four different future projections were made using the atmospheric $CO_2$ concentration pathways of RCP8.5, RCP6, RCP4.5 and RCP2.6 as provided by the CMIP5 [http://cmip-pcmdi.llnl.gov/cmip5/]. The simulations made with prescribed atmospheric $CO_2$ are subsequently called the concentration pathway (CP) simulations.

The future simulations were repeated using the $CO_2$ emission scenarios that were used by the Integrated Assessment Model to generate the future atmospheric $CO_2$ concentrations used in the RCPs. We subsequently refer to these simulations as the emission pathway (EP) simulations. EP simulations have prescribed atmospheric carbon emissions, and the atmospheric $CO_2$ is determined by considering how much carbon is absorbed by the land and ocean in our ESM. For each of the EP scenarios, the radiative forcing of non-$CO_2$ gases was converted into an equivalent $CO_2$ concentration and added to the simulated atmospheric $CO_2$ to maintain the same radiative forcing of non-$CO_2$ greenhouse gases as the corresponding CP simulation. This additional $CO_2$ was not seen by the land and ocean carbon modules. From the difference between the EP and CP simulations, we quantify the carbon-climate feedbacks, and we use these differences to investigate how carbon-climate feedbacks influence future atmospheric $CO_2$ concentration and ocean acidification.

The COAL simulations of the future carbon-climate feedbacks made with RCP8.5 and RCP2.6 were discussed by Zhang et al. (2014) with a focus on how the feedbacks increased warming due to reduced carbon uptake by the land. Zhang et al. (2014) showed the EP simulations had less than $0.4°C$ more global surface warming than the corresponding CP simulations by 2100. Here, we add the RCP4.5 and RCP6 scenarios and focus our study on ocean acidification.

In all our simulations, the vegetation scenario used by Lawrence et al. (2013) remained unchanged over the simulation period following the CMIP5 experimental design. We also neglected changes in anthropogenic N deposition over the simulation period, because of the large uncertainty in the future deposition rate and the small impact it has on net land carbon uptake (Zaehle et al., 2010).

To account for possible drift in the simulated climate and carbon pools, a control simulation with the atmospheric $CO_2$ held constant at 284.7 ppm was performed over the simulation period (1850 to 2100). Drifts in climate and carbon pool sizes were small (less than $0.015°C$ and 0.1 Pg C per century$^{-1}$), and correcting the future scenarios with the control simulation had negligible impact on the future projections of ocean acidification and ocean warming.

## 3 Results

### 3.1 Historical Period

An assessment of the simulated carbon and climate was made in Zhang et al. (2014), Matear and Lenton (2014), and Buchanan et al. (2016) and here we briefly comment on the simulation over the historical period (1850 - 2005). Mk3L-COAL simulates the historical climate well, as compared to the models used for earlier IPCC assessments (Phipps et al., 2011; Pitman et al., 2011). Over the historical period, the global averaged surface warms by $0.57°\pm 0.07°C$ (Zhang et al., 2014), which is comparable to the observed value of $0.76°\pm 0.19°C$ (Trenberth et al., 2007). The simulated land and ocean uptake were $85 \pm 1$ PgC and $116 \pm 1$ PgC, respectively, compared to observed land and ocean estimates of $135 \pm 84$ and $135 \pm 25$ PgC respectively(Zhang et al., 2014). The simulated responses of the land carbon cycle to increasing atmospheric $CO_2$ and warming is consistent with those from CMIP5 (Zhang et al., 2014), while the acidification of the ocean was also comparable to other CMIP5 simulations (Bopp et al., 2013).

For 2002, we compare the simulated annual mean surface ocean aragonite saturation state to the values estimated from GLODAPv2 observational dataset (Key et al., 2016; Olsen et al., 2016; Lauvset et al., 2016) (Figure 1). The simulated values are broadly consistent with the observations with the location of aragonite saturation value of 3 (purple line in Figure 1) being found at similar locations. However, the simulation slightly underestimates aragonite saturation state in the tropics.

### 3.2 Future Response

For the future, the ESM simulated higher atmospheric $CO_2$ in the EP simulations than the corresponding CP simulations and, by the end of the century, the atmosphere had 35, 60, 85 and 120 ppm more $CO_2$ in the EP simulations (RCP2.6, 4.5, 6, and 8.5 respectively) than in the corresponding CP simulations (Figure 2). This atmospheric $CO_2$ increase largely reflects more carbon being emitted to the atmosphere in the EP than the corresponding CP simulations. This is demonstrated by Zhang et al. (2014) who showed that for our ESM to track the atmospheric $CO_2$ concentration prescribed by RCP2.6 and RCP8.5, the emissions over this century need to be reduced by 69 and 250 Pg C, respectively. Therefore, the EP simulations have substantially more atmospheric $CO_2$ emissions than the corresponding CP simulations.

Since the differences in land and ocean carbon uptake between the corresponding EP and CP simulations reflect their different atmospheric $CO_2$ and global surface warming one way to interpret these differences is in terms of the feedback parameters of warming ($\gamma$) and elevated $CO_2$ ($\beta$) (Friedlingstein et al., 2006). Zhang et al. (2014) assessed the feedback parameters of our ESM using the 1% per year increase in atmospheric $CO_2$ simulations following the classical methodology of Friedlingstein et al. (2006).

For the ocean, our ESM feedback parameters ($\beta$=0.7 PgC/ppm, and $\gamma$=-7.4 PgC/K) are similar to the values from CMIP5 models ($\beta$=0.80 $\pm$ 0.07 PgC/ppm and $\gamma$=-7.8 $\pm$2.9 PgC/K) (Arora et al., 2013). With higher atmospheric $CO_2$ concentrations, the EP simulations have higher global averaged surface temperature (Figure 4) and increased oceanic uptake of $CO_2$ (Figure 3) than the corresponding CP simulations. With higher atmospheric $CO_2$ concentrations, more $CO_2$ is taken up by the ocean; this in turn reduces the oceans buffering capacity of $CO_2$, or Revelle Factor (Revelle and Suess, 1957), acting as a feedback to reduce ocean carbon uptake. These differential changes in buffering capacity across different scenarios explain why the changes in ocean carbon uptake are very similar. Ocean warming plays a small role in the change in oceanic uptake of $CO_2$ because the changes in warming between the EP and CP simulations are small and similar for all scenarios (Figure 4).

For the land, our ESM warming feedback ($\gamma$ = -34 PgC/K ) was within the range of the CMIP5 models ($\gamma$ = -58.4 $\pm$ 28.5 PgC/K) (Arora et al., 2013), however, the $CO_2$ feeedback ($\beta$= 0.18 PgC/ppm) was on the extreme low end of the CMIP5 model range ($\beta$=0.92 $\pm$ 0.44 PgC/ppm) (Arora et al., 2013). Our ESM is consistent with the two ESMs used in the CMIP5 analysis that had land carbon models with carbon and nitrogen cycles ($\beta$= 0.23 $\pm$0.01) (Arora et al., 2013). The $\beta$ value of a land carbon model is strongly reduced by nutrient limitation, because the land $CO_2$ fertilization effect saturates as the land carbon uptake becomes nutrient limited (Zhang et al., 2014). The simulated changes in land carbon uptake between the EP and CP simulations are small and similar for the various emissions scenarios. The similarity in the land uptake between scenarios reflects little difference in warming between the EP and CP simulations (Figure 4). However, the impact of the $\beta$ feedback appears in the RCP8.5 scenario as a stabilisation, and subsequent decline in the cumulative land uptake difference onwards from 2060 (Figure 4b).

A recent analysis of 11 ESMs of the RCP8.5 scenario (Friedlingstein et al., 2014) showed atmospheric $CO_2$ in 2100 would be 44 $\pm$ 97 ppm greater in the EP simulations than in the CP simulations. Our ESM simulated value is on the upper end of this range (120 ppm), but was consistent with one model used in the Friedlingstein et al. (2014), which included nitrogen cycle and had a similar $\beta$ land value.

The higher atmospheric $CO_2$ translated into higher $CO_2$ in the surface ocean. To quantify ocean acidification impacts, we show the aragonite saturation state values in the surface water for both the CP and EP simulations (Figure 5). Figure 5 illustrates how rising atmospheric $CO_2$ impacts the

carbon chemistry of the surface ocean. Two ways to quantify the ocean acidification in the surface water are to monitor where aragonite becomes chemically unstable or corrosive (aragonite saturation state of less than 1) and where aragonite saturation declines to less than 3, an approximate threshold for suitable coral reef habitat (Hoegh-Guldberg et al., 2007). In Figure 5, the white lines denote annual mean aragonite saturation state values of 1, and the purple lines the annual mean aragonite saturation state values of 3. A quick way to assess the ocean acidification impacts is by comparing how the white and purple lines differ between RCP scenarios (e.g. differences in a column) and how the carbon-climate feedbacks alters the surface chemistry changes (i.e. differences across a row). As one goes to higher future emissions scenarios (e.g. RCP2.6 to RCP8.5), the atmospheric $CO_2$ concentrations increase and the white lines move towards the equator and the surface area of the water in which aragonite is chemically unstable expands. In contrast, as one goes to higher emissions scenarios, the suitable regions for coral reefs shrink. In the RCP6 and RCP8.5 scenarios, there are no suitable coral reef regions by 2100 and a substantial portion of the polar Southern and Northern hemisphere have surface water corrosive to aragonite in agreement with previous studies (Ricke et al., 2013; Sasse et al., 2015). When the carbon-climate feedbacks are considered, there is a further expansion of aragonite under-saturated surface water, and a further reduction in the area suitable for coral reefs.

Figure 6 shows more clearly how carbon-climate feedbacks alter the rate of ocean acidification. The figure shows how the global surface area of aragonite under-saturation (a) and the global surface area of suitable coral reef habitat (b) change with time for the various scenarios. All EP scenarios show an acceleration in ocean acidification (dotted lines) compared to the corresponding CP simulations (solid lines).

For under-saturated aragonite surface water, the EP simulations all display a similar evolution to the corresponding CP simulations but with a more rapid onset of under-saturated conditions. For RCP8.5, the EP simulation leads the CP simulation by about 5 years. For RCP6, the EP simulation leads the CP simulation by about 10 years. For RCP4.5, the lead is nearly 20 years. While for RCP2.6, there is a similar 20-year lead in the emissions simulation but the area of under-saturated water is small due to the low atmospheric $CO_2$ which makes quantifying the lead uncertain. Further, in the RCP2.6 scenario, the atmospheric $CO_2$ starts to decline after 2050 (Figure 2) because the scenario has negative emissions in the second half of the century which enables some recovery in ocean acidification. Associated with the decline in atmospheric $CO_2$, is a reduction in surface ocean acidification (Figure 6b); hence, in this scenario there is a small reduction in the area of under-saturated water by the 2100 from the maximum value in the 2060s.

For all scenarios, the carbon climate feedbacks accelerate the onset of under-saturated aragonite conditions. However, it is in the mid to low emissions scenarios (RCP4.5 and RCP2.6) where the differences between EP and CP simulations are greatest and, hence, where the carbon-climate feedbacks are most significant.

For the surface ocean area suitable for coral reefs, the evolution of the EP simulations are similar to the corresponding CP simulations but, again, they lead the CP simulations. The more rapid onset of ocean acidification produces the largest difference in the RCP4.5 scenario where, by the end of the century, the suitable area for coral reefs in the EP simulation (18%) is less than half the CP simulation (37%). Under the high emissions scenarios (RCP6.0 and RCP8.5), there is no suitable habitat for coral reefs by 2100 with the time of disappearance occurring 15 and 6 years earlier in the EP simulations than in the CP simulations for RCP6 and RCP8.5, respectively. With the highest emission scenario (RCP8.5), there is such a large and rapid release of $CO_2$ to the atmosphere and ocean acidification impacts are so substantial that the differences between the EP and CP simulations are similar, but with a slight acceleration in the EP simulation.

The differences between the EP and CP simulations extend into the ocean interior. By 2100, the EP simulations show a shoaling of the aragonite saturation horizon (depth of where the aragonite goes under-saturated) than the corresponding CP simulations (Figure 7). For the RCP2.6, the difference is generally small because the rate of atmospheric $CO_2$ rise is weak and the penetration of carbon is not very different between the two EP and CP simulations. However, for the other emissions scenarios, the differences between the EP and CP simulations are substantial, particularly in the Southern Ocean and North Pacific (Figure 7b,c,d). Under the RCP4.5 scenario in the Southern Ocean, the EP simulated aragonite saturation horizon is more than 400 m shallower than the CP simulation. In this scenario, the surface water does not become under-saturated with respect to aragonite (Figure 5b) but the increase in ocean carbon uptake in the EP simulation is sufficient to significantly shoal the aragonite saturation horizon. Such a shoaling of the aragonite saturation horizon would have a detrimental impact on calcifying organisms such as pteropods inhabiting the Southern Ocean (Comeau et al., 2012). The RCP8.5 and RCP6 scenarios also display regions where the aragonite saturation horizon is more than 400 m shallower in the EP simulation than in the CP simulation. In both these scenarios, most of the Southern Ocean surface water is under-saturated with respect to aragonite (Figure 5c,d) and the largest shoaling occurs just outside of the Southern Ocean, where anthropogenic carbon taken up in the Southern Ocean is stored (Groeskamp et al., 2016). As more anthropogenic carbon is transported into the ocean interior in the EP simulations, it is the regions where the carbon is stored that show the greatest shoaling of the aragonite saturation horizon. The projected increased shoaling of the aragonite saturation horizon in the Southern Hemisphere with carbon climate feedbacks could be important to the future viability of deep water corals found in regions like south of Australia where living corals are generally confined to water above the aragonite saturation horizon (Thresher et al., 2011; Guinotte and Fabry, 2008).

## 4 Discussion

Here we employ an ESM to investigate the potential consequences of carbon-climate feedbacks on the future evolution of ocean acidification. With the emissions driven (EP) simulations, we show that carbon-climate feedbacks can significantly accerlate the future rate of ocean acidification. Therefore, accounting for carbon-climate feedbacks is important in projecting future ocean acidification impacts and trajectories.

The other salient point is carbon-climate feedbacks have the greatest impact under the medium to low emissions scenarios (RCP4.5 and RCP2.6). For the RCP4.5 scenario, the carbon-climate feedbacks nearly double the area of under-saturated surface water, and halve the area of surface water suitable for coral reefs by the end of the century. While less dramatic, in the RCP2.6 scenario, the carbon-climate feedbacks reduce the area suitable for coral reefs by 40% and increase the area of under-saturated surface water by 20%. If we aim to track a low emissions scenario (Anderson and Peters, 2016), then we are on a path where the carbon-climate feedbacks can have the greatest impact on ocean acidification and there is a pressing need to better quantify the carbon-climate feedbacks to ensure models properly project the future ocean acidification. If we want to minimise ocean acidification impacts, we may require faster reductions in $CO_2$ emissions and we may need to consider ways to increase negative emissions (Lackner, 2016). Here is another area where ESM simulations can help assess the benefits and consequences of different strategies to enhance carbon sinks (Keller et al., 2017).

Here, we have only considered ocean acidification impacts, but carbon-climate feedbacks also lead to faster global warming. This would accelerate impacts like ocean warming and deoxygenation (Cocco et al., 2013). For our simulations, the carbon-climate feedbacks on these impacts were small (e.g. global ocean surface water less than 0.4 °C warmer), but these impacts are synergistic (Bopp et al., 2013) and they will further stress the ocean ecosystems with potential consequences for the future livelihood of coastal nations (Mora et al., 2013). Repeating future climate and ocean acidification impact assessments with ESM simulations that consider carbon-climate feedbacks is required to more realistically quantify the future changes in the ocean. As aragonite saturation state is also controlled by temperature (Mucci, 1983), there is a weak increase in saturation state with increased ocean warming, but this effect is very small in our ESM simulations and cannot offset the decrease in saturation state due to enhanced ocean carbon uptake.

### 4.1 Variability within scenario from multiple simulations

While it is natural to compare the impact of climate-carbon feedbacks on ocean acidification to previous estimates of intermodel variability from CP simulations, we emphasise, all the CP simulations prescribe the future atmospheric $CO_2$ concentrations. However, to broaden the discussion, we review the current results of intermodel differences for ocean acidification. Bopp et al. (2013) provided a

seminal study of the intermodel variability in the projected ocean acidification from CP simulations with different emissions scenarios. For the four emission scenarios considered here, Bopp et al. (2013) showed the global change in surface aragonite saturation state had a small intermodel range (less than 10%) and they concluded that atmospheric $CO_2$ dominated the model behaviour. This is consistent with Hewitt et al. (2016) who observed similar behaviour in the ocean carbon responses between CMIP5 projections for a given scenario.

Regionally, Bopp et al. (2013) showed significant intermodel differences that are comparable to the magnitude of the carbon-climate feedbacks we simulated. For example, in the Southern Ocean (South of 60°S), Bopp et al. (2013) showed mean aragonite saturation of the surface water occurs in 2067 and 2092 for RCP8.5 and RCP6.0, respectively, but from the intermodel variability it could occur 7 and 13 years earlier for RCP8.5 and RCP6, respectively. While the intermodel differences are large, Séférian et al. (2016) showed that much of these regional differences is attributed to differences in spin-up protocol, which influence both a model simulated pre-industrial state and the amount of drift that occurs in the subsequent scenario simulation. Thus Séférian et al. (2016) suggests that much of Bopp et al. (2013) estimated model-model uncertainty reflects inconsistencies in spin-up protocol and initial conditions across CMIP5 ESM simulations rather than how these models parameterise key biogeochemical processes. Further, if the carbon-climate feedbacks increase the projected atmospheric $CO_2$ then, all these models would simulate an earlier onset of aragonite saturation in the surface water. Hence, our study complements Bopp et al. (2013) model-model analysis by introducing an independent modification to their analysis associated with carbon-climate feedbacks. We next assess the robustness of our simulated carbon-climate feedbacks by comparing our ESM to other CMIP5 models.

## 4.2 Robustness of the simulated Carbon-Climate Feedbacks

The World Climate Research Program (WCRP) identified *Carbon Feedbacks in the Climate System* as one of their Grand Challenges (https://www.wcrp-climate.org/grand-challenges/gc-carbon-feedbacks) due to the potential influence the feedbacks may have on future climate change (Jones et al., 2013). A key conclusion of our study is that the carbon-climate feedbacks may also be important to the future trajectory of ocean acidification. Our estimates of the impact of the carbon-climate feedbacks on ocean acidification are only based on a single model and, to help assess the robustness of our results we compare our ESM with other CMIP5 simulations. To compare our ESM to other CMIP5 models, we compare the land and ocean feedback parameters of warming ($\gamma$) and elevated $CO_2$ ($\beta$) (Friedlingstein et al., 2006).

Zhang et al. (2014) assessed the feedback parameters of our ESM and showed our ocean response was consistent with CMIP5 models (Arora et al., 2013). Nutrient limitation had the greatest impact on our ESM land $\beta$ feedback parameter, which significantly reduced land carbon uptake with rising atmospheric $CO_2$. Hence, in emission scenarios with high atmospheric $CO_2$ concentrations

(RCP8.5) our ESM had less land carbon uptake and higher atmospheric $CO_2$ than most CMIP5 models (Friedlingstein et al., 2014). While this behaviour is on the extreme end of the CMIP5 models it partially reflects the omission of nutrient limitation in most CMIP5 land models, and the models that do include nutrient limitation have $\beta$ values similar to our ESM.

Recent studies show the carbon-climate feedbacks are dominated by the land carbon cycle response to warming (Jones et al., 2013; Boer and Arora, 2013; Hewitt et al., 2016). From the CMIP5 simulations, for the historical period, the land $\gamma$ was -49 $\pm$ 40 PgC K$^{-1}$ (Wenzel et al., 2014). For comparison, the ESM used in this study the land $\gamma$ for the historical period was -35 PgC K$^{-1}$, which is within the range of the CMIP5 models. Recent analysis using the short-term variability to further constrain the model simulations reduces the range of the land $\gamma$ to -44 $\pm$ 14 PgC K$^{-1}$ (Wenzel et al., 2014) Our ESM again falls within this reduced range but, when compared to the weighted mean of the CMIP5 models, our ESM is at the lower range of these estimates.

An ESM with a weaker land $\gamma$ feedback parameter equates to lower atmospheric $CO_2$ concentrations in future projections. Therefore, in the scenarios with relatively low atmospheric $CO_2$ concentrations (RCP2.6 and RCP4.5), where the land $\beta$ feedback is small and the land $\gamma$ feedback dominates, our ESM is on the low side of the CMIP5 models and could be providing a lower bound estimate of the carbon-climate feedbacks on future ocean acidification.

## 5   Conclusions

The large differences in the carbon-climate feedbacks are not only a key uncertainty in climate projections (Jones et al., 2016; Friedlingstein et al., 2014) they are also a key uncertainty in future ocean acidification projections. The future response of the land carbon uptake may be further reduced by coupling between increasing climate extremes and induced $CO_2$ losses to the atmosphere (Reichstein et al., 2013), which is poorly represented in ESM simulations. Therefore, for both climate projections and ocean acidification, there is a pressing need to improve our ability to simulate the carbon-climate feedbacks and the C4MIP simulations (Jones et al., 2016) will be crucial for better quantifying the future impact of the carbon-climate feedbacks on ocean acidification. However, it is important the C4MIP simulations give proper consideration to the initialisation and spin-up of the carbon cycle (Séférian et al., 2016), because we will want to use these simulations to assess regional rates of ocean acidification. Even with the small carbon-climate feedbacks shown here (less than 120 ppm change by the end of the century), similar to CMIP5 model range, the impact on the future rate of ocean acidification is still significant and makes the ocean more vulnerable than what was provided by the recent ocean acidification assessment (Aze et al., 2014).

*Acknowledgements.* The research was funded by the Australian Climate Change Science Program and the CSIRO Oceans and Atmosphere.

*Competing Interests.* The authors declare that they have no competing financial interests.

*Correspondence.* Correspondence and requests for materials should be addressed to R J Matear (email: richard.matear@csiro.au).

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

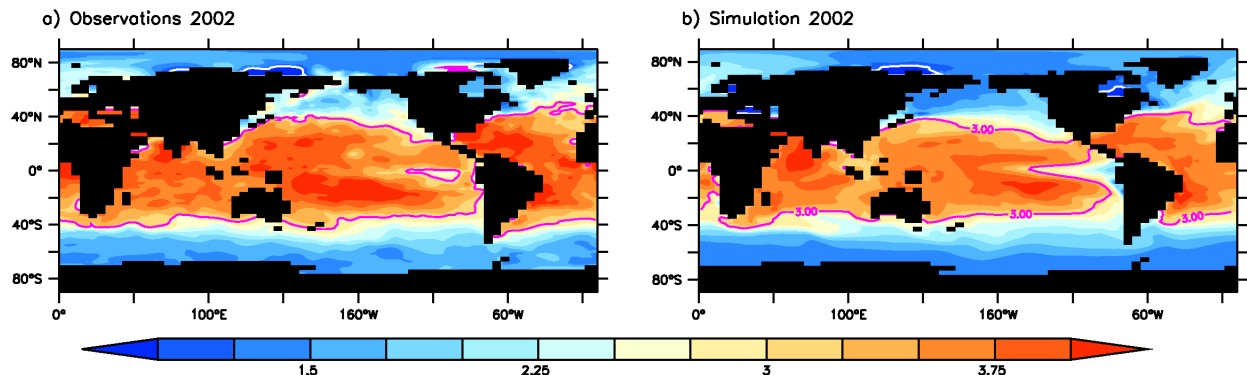

**Figure 1.** 2002 Surface aragonite saturation state a) from the GLODAPv2 observational dataset (Key et al., 2016; Olsen et al., 2016; Lauvset et al., 2016) and b) simulated. The purple line denotes an aragonite saturation state of 3.

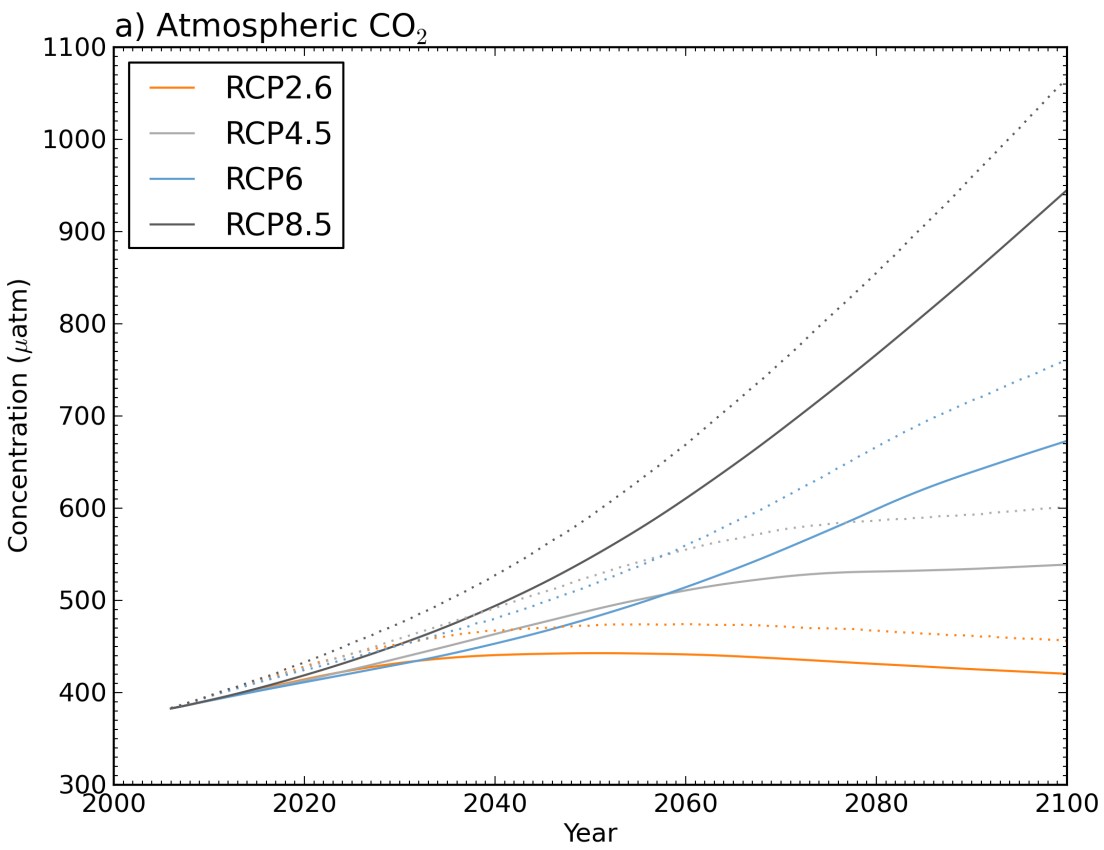

**Figure 2.** For the various RCP scenarios, the atmospheric $CO_2$ prescribed for the CP simulations (solid lines) and simulated by the EP simulations (dotted lines).

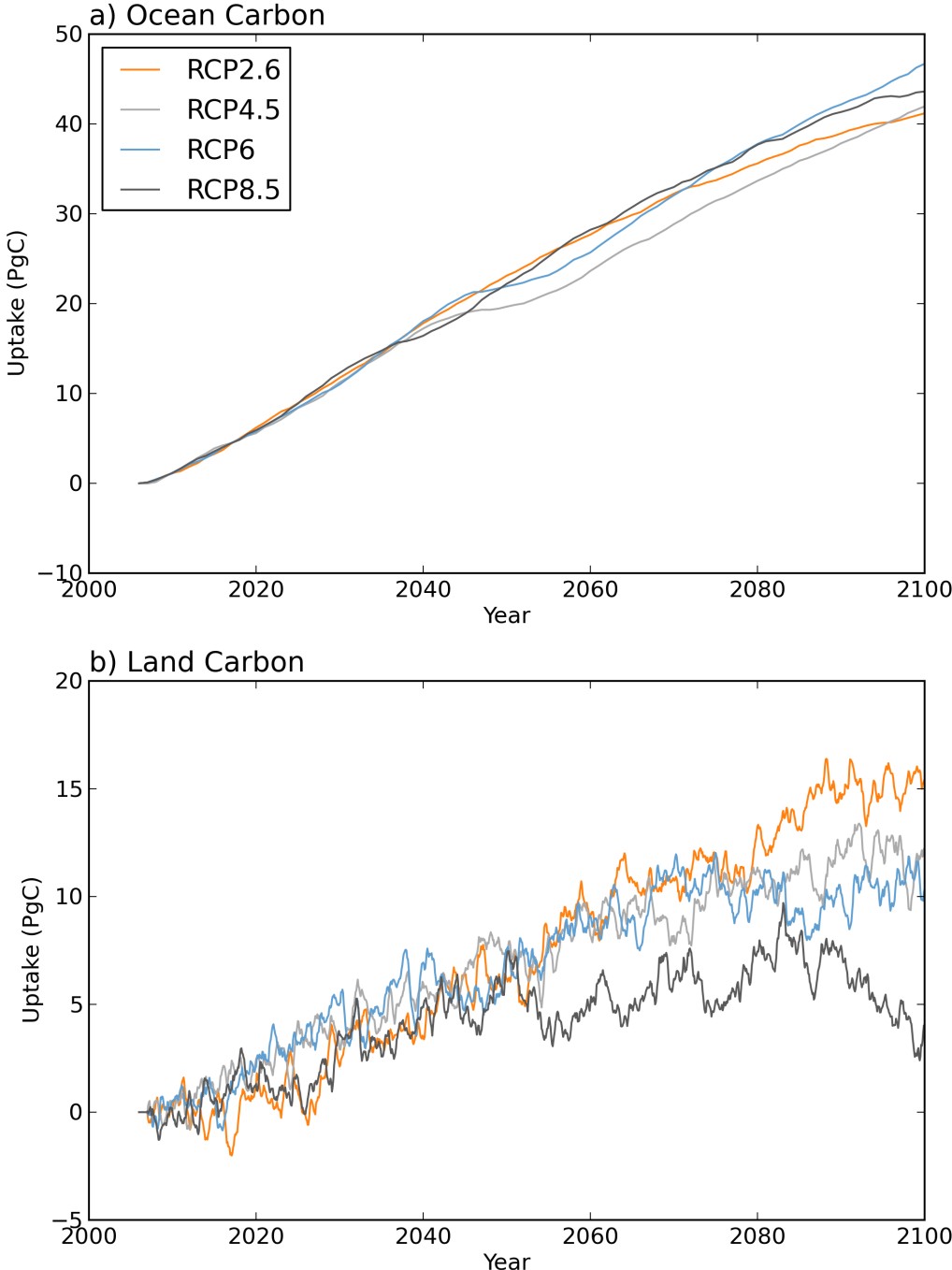

**Figure 3.** For the various RCP scenarios, the cumulative difference in a) ocean carbon uptake (PgC) and b) land carbon uptake (PgC ) between the EP and corresponding CP simulation.

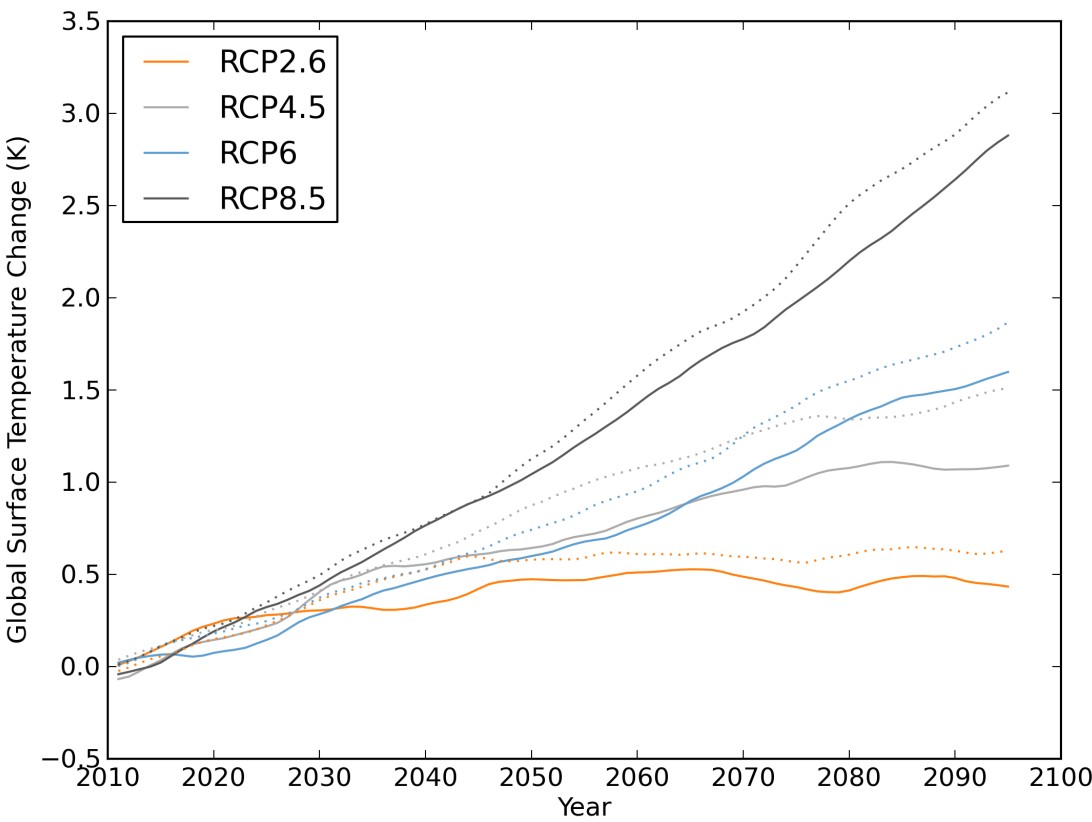

**Figure 4.** For the various RCP scenarios, the global decadal averaged surface temperature change from the present-day for the EP (dotted) and CP (solid lines) simulations.

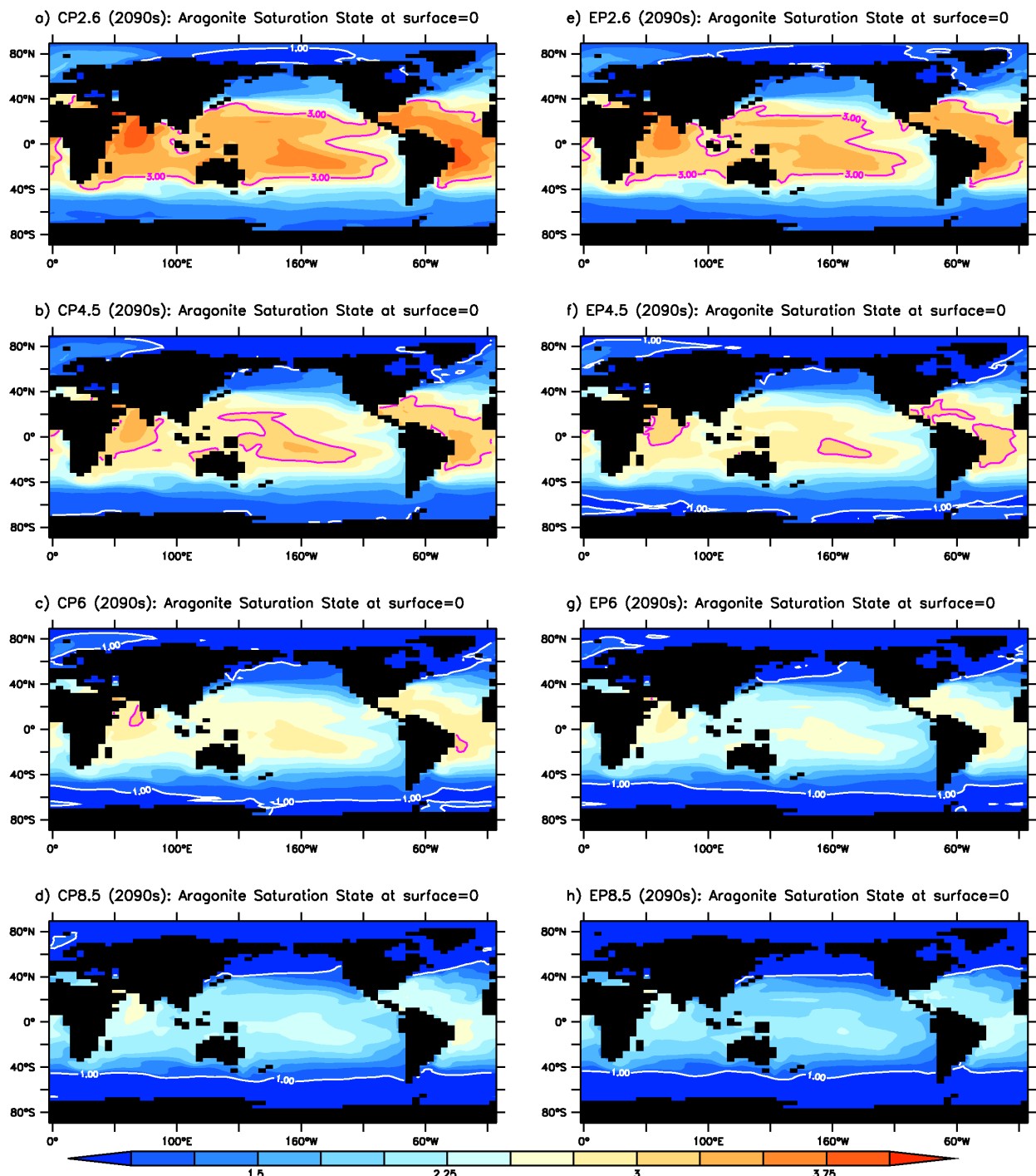

**Figure 5.** For the various RCP scenarios, the surface ocean aragonite saturation state for the decade of the 2090s for the CP simulations (left column): a) RCP2.6, b) RCP4.5, c) RCP6, and d) RCP8.5; the EP simulations (right column) e) RCP2.6, f) RCP4.5, g) RCP6 and RCP8.5. In the figures, the white contour lines denote where aragonite saturation state equals one. The purple contour lines denote aragonite saturation state of 3.

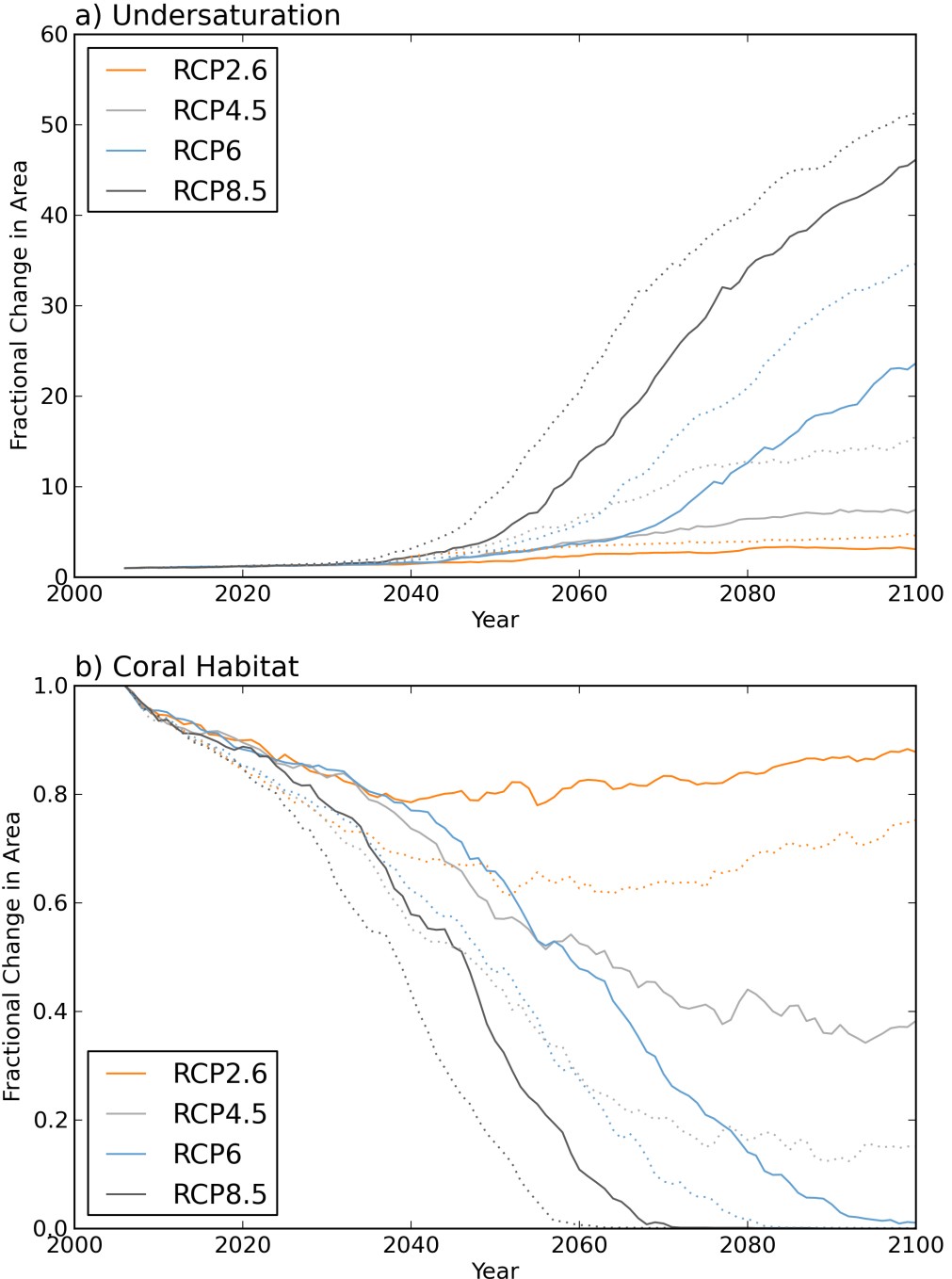

**Figure 6.** For the various RCP scenarios, the CP simulations (solid lines) and their corresponding EP simulations (dotted lines) for: a) Change in area of surface water with aragonite saturation state less than 1 relative to the area in 2005; b) Change in area of the surface water suitable for coral reefs (aragonite saturation state greater than 3) relative to the area in 2005.

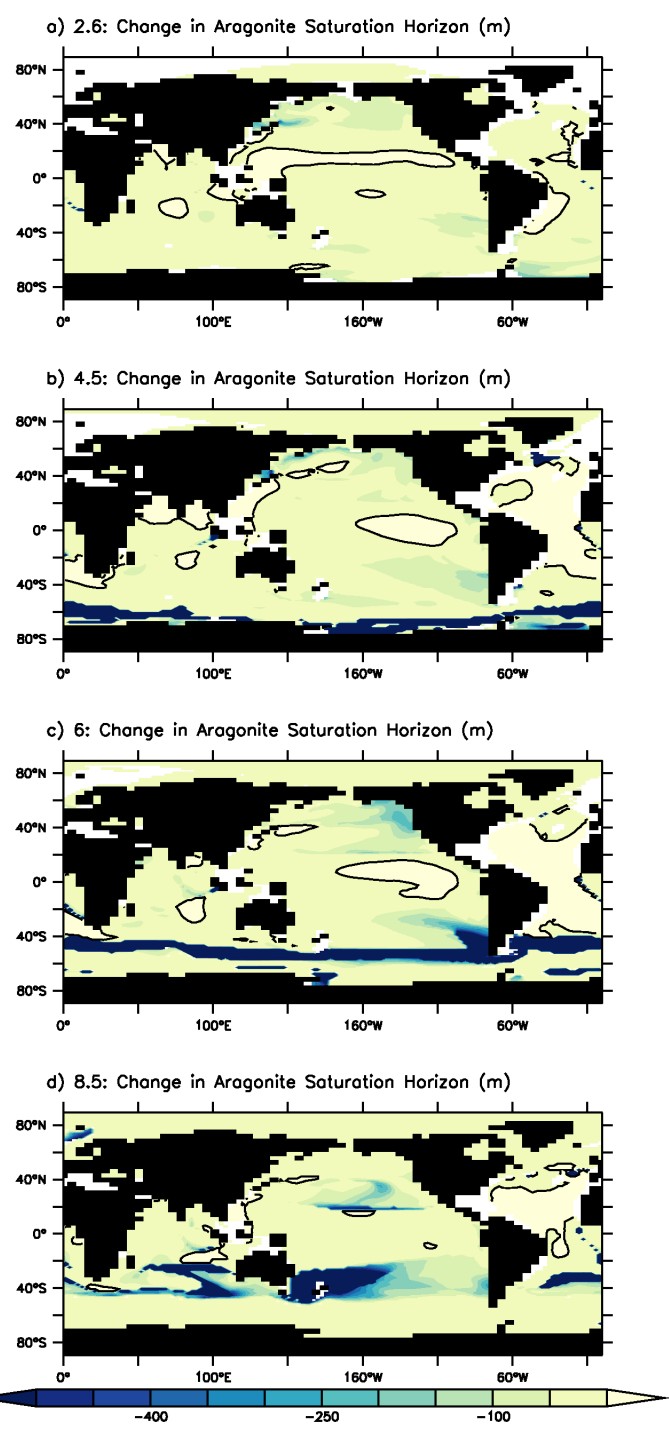

**Figure 7.** For the year 2100, the change in the depth of the aragonite saturation horizon between the emission simulations (EPs) and the concentration simulations (CPs) for: a) RCP2.6; b) RCP4.5; c) RCP6; and d) RCP8.5