# Peer review of "Carbon-climate feedbacks accelerate ocean acidification"

_Biogeosciences, 2017_

## Referee Comment (RC1) · Anonymous Referee #1 · 24 Jul 2017

"Sensitivity of future ocean acidification to carbon climate feedbacks" by Matear and Lenton discusses how much the rate of ocean acidification can be accelerated by the carbon-climate feedbacks. In general, this problem is very complex however the authors have chosen to discuss one of its most straightforward angles: what is the difference between ocean acidification characteristics in a given model which is run with and without carbon-climate feedbacks.

For the conclusions to be meaningful for the wider community (outside of the circle of developers of this particular model) and before going into the details of the actual numbers with implications to e.g. corals and potential significance to global carbon policy, two aspects need to be clarified:

1. Carbon-Climate feedbacks are one of the key uncertainties in future climate projec-

tions (the authors comment on this fact a few times across the manuscript). How does the strength of the carbon-climate feedbacks in this particular model compare with that of the other CMIP5 models? EP vs CP is a standard protocol, so data is available. To which part of the model spread do you belong to?

2. Not taking into account carbon-climate feedbacks in acidification projections introduces some uncertainty as was demonstrated by the authors. How does this uncertainty compare to other sources of uncertainty in projections of ocean acidification? Shall we (in extreme case) dismiss a study of e.g. Bopp et al 2013 or any concentration-forced ocean model projections which might be superior in some other aspects of parameterising ocean acidification?

I would strongly encourage the authors to address the issues above before resubmitting the manuscript.

Some other comments:

Zaehle et al, 2010 is referred to for the carbon-climate feedback significance; however this paper is dedicated to the terrestrial nitrogen feedbacks. Is this a good reference? More generally, the manuscript will benefit from more detailed review of the literature on the significance on carbon-climate feedbacks.

l.38-42: you describe only one of the climate-carbon feedbacks but then refer to it in plural. Could you give description of other climate-carbon feedbacks? See also my previous comment.

l.145: actually the figure shows a systematic overestimate of carbonate ion. A more realistic assessment of model skill is needed.

Abstract and throughout the text re: More substantial significance of the carbon-climate feedbacks in lower emission scenarios. I would encourage the authors to use more careful language and to alert the reader that this relates to the relative impact rather than absolute impact. I am afraid this message might be taken out of the context and

subsequently misinterpreted.

Abstract: the last sentence does not make sense. Please rephrase more accurately.

Throughout the paper: Could you please proofread the manuscript? I noticed a lot of misspelled words, missing prepositions, and excessive use of "this" (occasionally 3-4 times in a single sentence). Not being a native English speaker myself, I will not go into listing all the occurrences of poor grammar I have spotted, but I would like to bring this to the editor's attention.

---

## Referee Comment (RC2) · Anonymous Referee #2 · 29 Jul 2017

This manuscript describes how ocean acidification may be accelerated when carbon-climate feedbacks are accounted for in their model. This is a very complex and important topic which is interesting to a large scientific community. However, since the authors here take a relatively simple approach and use only one model the impact is limited. In my opinion the manuscript should be published only after some revisions, as outlined below.

Major issues: There is not sufficient discussion of the difference between the dynamics and responses of the carbon sinks between the EP and CP simulations. Nor how the sink responses in your particular model compares to those in other models. There is not sufficient discussion of the changes in ocean and land uptake in the EP simulations compared to CP simulations. Especially I would like to see more discussion about the

[Figure]

overall effect these changes have on the atmospheric CO2. Again, comparison with other models would be helpful. Line 108-109: This sentence implies to me that the results are highly dependent on the land and ocean biogeochemistry in the model, and the dynamics of these sinks. The results will therefore be very model-dependent and some more discussion about how representative your particular model is, is necessary. Line 143: You use 1995 as a reference year here. Why 1995 and GLODAPv1 when there is much more recent data for 2002 and GLODAPv2? Lines 153-155: The statement here implies that the increase in atmospheric CO2 concentration in the EP simulations are due to reduced land uptake. How does that fit with Figure 4? A better explanation is necessary here. Line 184: Here it is stated that the EP scenarios are more negative than the CP simulations. But on Figure 6 all numbers are positive. Rephrase. Line 160 and Figure 4: Firstly, what is the reference year here? Secondly, Is the caption for the figure correct? The label on the y-axis says PgC which suggests that this is a cumulative difference, but the caption states that this is the annual difference (in PgC yr-1). The numbers are very large given the small difference in atmospheric CO2 and temperature between the EP and CP simulations. Please clarify. In my opinion this is one of the most important figures in the manuscript so a more thorough description of it and discussion of the results shown is necessary.

Minor points: The reference to Lenton et al (2015) is wrong. This is the Biogeosciences Discussions version but there is a peer-reviewed version from 2016. In the Figure 1 caption it should read "carbonate ion". In the Figure 3 caption the year defined as present-day needs to be defined and stated. Please proof-read the entire manuscript carefully. There are many instances of misspelled words, and quite a bit of poor phrasing detracts from the reading.

---

## Author Comment (AC1) · 13 Oct 2017

We thank the reviewer for the constructive comments. The following are our point by point response to these comments.

1. Carbon-Climate feedbacks are one of the key uncertainties in future climate projections (the authors comment on this fact a few times across the manuscript). How does the strength of the carbon-climate feedbacks in this particular model compare with that of the other CMIP5 models? EP vs CP is a standard protocol, so data is available. To which part of the model spread do you belong to

Response: We added a more through comparison of both the CP and EP driven climate projections to the discussion (new section included). The key point is our simulations fall within the range of previous projections, To show this we include recent analysis from Arora's et al 2013 ( J of Climate)

2.Not taking into account carbon-climate feedbacks in acidification projections introduces some uncertainty as was demonstrated by the authors. How does this uncertainty compare to other sources of uncertainty in projections of ocean acidification? Shall we (in extreme case) dismiss a study of e.g. Bopp et al 2013 or any concentration-forced ocean model projections which might be superior in some other aspects of parameterising ocean acidification?

R: We apologise for any lack of clarity. To put this paper in context, the study is complementary to previous published studies such as Bopp et al (2013). Here, we argue that by including carbon-climate feedbacks that the atmospheric concentration used to drive the model is higher than in the atmospheric concentration driven run. The implication of this is that the magnitude of the changes in ocean acidification would be larger (by the end of the century) and occur sooner than previously reported. The uncertainty estimated from projections using concentration driven simulations should not change significantly, rather a positive carbon-climate feedback would accelerate ocean acidification in all these simulations.

To reflect these comments, we have added these points to the discussion

1. the manuscript will benefit from more detailed review of the literature on the significance on carbon-climate feedbacks.

R: as part of revising the paper to address the general points above we have added several references on the carbon-climate feedbacks included in the C4MIP special collection (http://journals.ametsoc.org/topic/c4mip) such as Arora et al. 2013, and Boer and Arora 2013. We also included Jones et al. 2016 CMIP6 paper and Reichstein et al. 2013 (Nature)

l.38-42: you describe only one of the climate-carbon feedbacks but then refer to it in

plural. Could you give description of other climate-carbon feedbacks? See also my previous comment.

R: Thank you we have now expanded our discussion of the potential feedbacks

l.145: actually the figure shows a systematic overestimate of carbonate ion. A more realistic assessment of model skill is needed.

R: We have now added a more rigorous assessment of model skill which includes a comparison with other published results of model projections.

Abstract and throughout the text re: More substantial significance of the carbon-climate feedbacks in lower emission scenarios. I would encourage the authors to use more careful language and to alert the reader that this relates to the relative impact rather than absolute impact. I am afraid this message might be taken out of the context and subsequently misinterpreted.

R: We agree with the statement and we have tried to be more careful in discussing the relative and absolute changes. We agree, the high emission scenario produces dramatic OA changes.

Abstract and rest of paper

R: we have carefully proof read and edited the paper to reduce the occurrence of poor grammar.

---

## Author Comment (AC2) · 13 Oct 2017

We thank the reviewer for the constructive comments. The following are our point by point response to these comments.

Major issues:

1. There is not sufficient discussion of the difference between the dynamics and responses of the carbon sinks between the EP and CP simulations. Nor how the sink responses in your particular model compares to those in other models.

Response: We agree this is an important point to address, we now devote a new section in the paper comparing our model simulations to previously published results on carbon climate feedbacks with the different emissions scenarios. We have also

compared our results to previous CP simulations. (Please see response to Reviewer 1)

2. There is not sufficient discussion of the changes in ocean and land uptake in the EP simulations compared to CP simulations. Especially I would like to see more discussion about the overall effect these changes have on the atmospheric CO2. Again, comparison with other models would be helpful.

R: We agree that it is important to show how the carbon-climate feedbacks influence land and ocean carbon uptake. We have added figures showing land and ocean uptake changes and compare our simulations to previously published results, please see Reponse to Reviewer 1.

3. Line 108-109: This sentence implies to me that the results are highly dependent on the land and ocean biogeochemistry in the model, and the dynamics of these sinks. The results will therefore be very model-dependent and some more discussion about how representative your particular model is, is necessary.

R: In fact, this is true for the entire Earth System and indeed the point of the paper, by prescribing atmospheric concentrations of CO2 the magnitude of sink and sources are irrelevant i.e. the response of the ocean chemistry is to atmosphere and not the changing sinks. However, to provide confidence for the reader we have added a new section in the discussion comparing our simulations to previously published simulations. Our simulated carbon-climate feedback falls within the range of previous results but some models show a much greater feedback which would give a greater impact on OA than presented here.

Line 143: You use 1995 as a reference year here. Why 1995 and GLODAPv1 when there is much more recent data for 2002 and GLODAPv2?

R: We have changed the dataset to GLODAPv2 referenced to Lauvset et al (2016), ESDD

Lines 153-155: The statement here implies that the increase in atmospheric CO2 concentration in the EP simulations are due to reduced land uptake. How does that fit with Figure 4? A better explanation is necessary here.

R: We have added a figure of the land uptake, and modified the text to make this point clearer. With reduced uptake by the land the atmospheric CO2 levels increase which increases uptake by the ocean but it is less than the land response.

Line 184: Here it is stated that the EP scenarios are more negative than the CP simulations. But on Figure 6 all numbers are positive. Rephrase.

R; fixed and rewrote

Line 160 and Figure 4: Firstly, what is the reference year here? Secondly, Is the caption for the figure correct? The label on the y-axis says PgC which suggests that this is a cumulative difference, but the caption states that this is the annual difference (in PgC yr-1). The numbers are very large given the small difference in atmospheric CO2 and temperature between the EP and CP simulations.

R: The caption was incorrect - the figure shows cumulative uptake. In addressing the previous points, we have added figures of land and ocean uptake to clearly show how the land and ocean respond in the EP simulation.

Minor points:

The reference to Lenton et al (2015) is wrong. This is the Biogeosciences Discussions version but there is a peer-reviewed version from 2016.

R: fixed

In the Figure 1 caption it should read "carbonate ion".

R: Fixed

In the Figure 3 caption the year defined as present-day needs to be defined and stated.

R:Yes - now 2002 based on GLODAPv2

Please proof-read the entire manuscript carefully.

R: The manuscript has been rewritten and minor errors addressed